# Recognition and Diagnosis of Underlying Disease Processes in Bacterial Pneumonia

**DOI:** 10.3390/ani14111601

**Published:** 2024-05-29

**Authors:** Jonathan D. Dear, Sean E. Hulsebosch, Lynelle R. Johnson

**Affiliations:** Department of Medicine and Epidemiology, School of Veterinary Medicine, University of California, Davis, CA 95616, USA; shulsebosch@ucdavis.edu (S.E.H.); lrjohnson@ucdavis.edu (L.R.J.)

**Keywords:** pneumonia, aspiration, bacteria, dog, cat

## Abstract

**Simple Summary:**

Bacterial pneumonia is an important and common diagnosis in small animals with respiratory disease. This review discusses the clinical presentation and diagnostic approach for dogs and cats suspected of having pneumonia.

**Abstract:**

Bacterial pneumonia can present in both dogs and cats as either acute or chronic disease, and clinical signs may suggest respiratory or systemic impairment. Aspiration pneumonia, a common form of disease, can be caused by esophageal or laryngeal disease, vomiting, or altered mentation, such as with CNS disease or general anesthesia. Bacterial pneumonia can also develop due to inhaled airway foreign bodies, reduced local immune function, or impaired clearance of respiratory secretions. Rapid identification of the etiology and associated risk factors for pneumonia can help veterinarians develop a targeted therapeutic plan and prevent recurrence of disease.

## 1. Introduction

Dogs and cats can be diagnosed with acute, fulminant bacterial pneumonia or may present with chronic, low-grade pneumonia with non-specific signs of illness. Typical respiratory abnormalities, including cough, tachypnea, and respiratory effort, might be accompanied by signs of systemic illness, such as fever, malaise, and decreased appetite. Identification of underlying systemic diseases associated with the development of bacterial pneumonia can assist in obtaining an accurate diagnosis. The rapid identification of clinical findings suggesting bacterial pneumonia is essential for prompting appropriate intervention. Likewise, selecting appropriate diagnostic tests to determine the underlying etiology, as well as the severity of illness, is critical to guide therapy.

## 2. Definition and Etiology

Pneumonia is defined as an infection of the lungs affecting the alveolar epithelial lining and can be caused by viruses, bacteria, protozoa, fungi, and parasites. Of these, the most common and widespread etiology in dogs and cats is likely bacterial pneumonia, which can be primary or secondary. Identifying the predisposing disease processes for secondary pneumonia can be challenging. In human medicine, bacterial pneumonia is often categorized as community-acquired, hospital-acquired, healthcare-associated, or ventilator-associated pneumonia; however, these classifications are not readily applicable to veterinary medicine. Dogs and cats can develop infectious pneumonia caused by primary respiratory pathogens, such as *Bordetella* or *Mycoplasma* spp., by particularly virulent organisms that overwhelm defense mechanisms, due to aspiration pneumonia or through foreign body inhalation [1,2,3]. Conditions that can predispose to bacterial pneumonia include local immune deficiency from ciliary dyskinesia or IgA deficiency, or from bronchiectasis [4,5,6]. Underlying systemic disorders that can predispose to bacterial infections such as pneumonia include diseases associated with defective neutrophil function, such as Cushing’s syndrome or diabetes mellitus, and systemic viral infections in cats [7,8]. In these situations, pneumonia can occur because organisms are able to penetrate and infect the lung parenchyma, overcoming host defense mechanisms. Although iatrogenic immunosuppression from chronic glucocorticoid use is a potential risk factor for pneumonia, it is rare in clinical practice to identify bacterial pneumonia in animals treated with oral glucocorticoids for diseases such as inflammatory airway disease or immune-mediated systemic disease.

Pneumonia can be difficult to distinguish from pneumonitis, a sterile inflammatory condition of the alveolar lining that can precede pneumonia in some cases [9,10]. The classic example of pneumonitis has been described in people as a chemical injury to the lungs caused by gastric acid aspiration, in which gastric acid suppresses bacterial growth in the aspirant. This chemical injury can induce bronchoconstriction, ventilation:perfusion mismatching, and acute respiratory distress syndrome [10]. In addition, pneumonitis likely predisposes the alveolar epithelial lining to subsequent bacterial invasion, and this presumption has resulted in the widespread use of antibiotics in animals that suffer aspiration injury, even when evidence of infection is not established. Aspiration events commonly result in a clinical diagnosis of pneumonia rather than pneumonitis because manifestations of disease are acute and severe, particularly in cats, although this species is affected less commonly than dogs by aspiration [11,12,13]. An additional clinical concern is that non-invasive indicators of disease, such as hematology, diagnostic imaging, and biomarkers, might not accurately discriminate between pneumonia and pneumonitis [9,14].

Pneumonia can occur as a primary event due to exposure to infectious organisms or inhalation of a virulent species, but it more commonly develops as a secondary condition. Aspiration pneumonia is often suspected in animals that have vomiting associated with an enteropathy or systemic disease, or regurgitation due to esophageal dysfunction or stricture [2,15]. Other common causes of aspiration include laryngeal dysfunction, neurologic disorders associated with seizures or recumbency, and general anesthesia [2,16,17]. Dogs with laryngeal paralysis (and particularly those resulting from polyneuropathy) are at high risk of aspiration events with or without laryngeal lateralization surgery. Aspiration pneumonia is particularly prevalent in brachycephalic breeds due to the high incidence of hiatal herniation, aberrant pharyngeal anatomy, and esophageal dysmotility that results from chronic upper airway obstruction. Aspiration events are important not only because of the development of pneumonia but also because chronic aspiration can result in substantial airway injury with the formation of bronchiectasis, which can then lead to recurrent infection due to the trapping of secretions in the lower airways [18,19]. Other causes of bronchiectasis that can predispose to recurrent pneumonia include congenital disease, inhalational or toxic airway injury, inflammatory airway disease, airway foreign bodies, and primary ciliary dyskinesia [4,11,20,21,22].

In certain parts of the world, bacterial pneumonia associated with foreign body inhalation is commonly reported [23,24]. In particular, areas that have a proliferation of grass awns in the dry season and a large population of field trial or hunting dogs will commonly encounter this type of secondary pneumonia. This condition is less common in cats than in dogs, likely because of their smaller airways and lifestyle differences from dogs. Given the non-standard geographic distribution of grass awns in the environment, evaluating the patient’s history for potential exposure is important in considering this differential diagnosis for pneumonia.

Primary ciliary dyskinesia results from a heritable defect in microtubule formation that is likely the result of autosomal recessive inheritance in certain pure-bred dogs (e.g., Old English Sheepdogs, Newfoundlands, and Alaskan malamutes) [5,25,26]. This disorder results from mutations in the genes responsible for dynein arm structure and function in cilia throughout the body. It can be associated with deafness, hydrocephalus, reproductive failure, and, in the respiratory tract, leads to failure of mucociliary clearance. The typical presentation is that of antibiotic-responsive nasal discharge and cough in a young, purebred dog. In humans, approximately 50% of ciliary dyskinesia patients will display situs inversus and 25% of those will have bronchiectasis early in life [27]. This triad of findings is referred to as Kartagener’s syndrome and has been occasionally reported in dogs. In the clinical syndrome of primary ciliary dyskinesia, some dogs will have recurrent respiratory signs that lessen or resolve over time while others develop bronchiectasis and life-threatening recurrent infections [5]. Primary ciliary dyskinesia is much less common in cats than in dogs and no specific breed predilection has been reported. Secondary ciliary dyskinesia can develop as a consequence of chronic airway infection or inhalation injury, but the incidence and outcomes of this condition are poorly described in the veterinary literature [28].

Systemic immunosuppression as a risk factor for bacterial pneumonia appears to be more common in cats than in dogs, with up to 70% of cats and kittens affected by a systemic immunosuppressive condition in one study of fatal feline pneumonia [12]. Infection with FeLV and/or FIV was most common in that study, but other chronic diseases, such as diabetes mellitus and chronic kidney disease, potentially contributed to immunosuppression in older cats. The association with systemic viral infection has not been borne out in large scale studies of pneumonia but could be an important consideration, given the prevalence of viral disorders in the cat population.

Viral pneumonia can precede bacterial pneumonia and play a role in reducing host defenses, such that secondary bacterial colonization becomes more efficient. Several studies have found that 13–30% of dogs with bacterial pneumonia had concurrent evidence of parainfluenza virus infection [29,30] and secondary bacteria were involved in 32% of fatal feline pneumonia cases associated with viral causes [12]; however, large-scale studies are lacking. Ancillary tests, such as viral culture or PCR testing, can help to document concurrent infections or exposure to viral organisms; however, there are few peer-reviewed studies evaluating concurrent infections in the pneumonia of dogs and cats.

## 3. Signalment and History

The history in an animal with pneumonia typically includes cough, but can also involve increased respiratory rate or effort, gagging, or retching. Loss of appetite and weight loss can be found in more severely affected animals with chronic disease, and other aspects of the history can provide clues to the underlying cause of pneumonia [11,15]. Exposure to other affected dogs or cats should raise suspicion for an infectious cause of pneumonia, perhaps initiated by a virus and then perpetuated by bacterial invasion. Infectious respiratory pneumonia is typically encountered 3–10 days post-exposure to other dogs and cats and, while most common in younger animals, it should not be discounted in mature, well-vaccinated individuals. This etiology seems to be more common in dogs, while cats share infectious upper respiratory tract diseases more commonly than lower respiratory tract infections.

Pneumonia related to foreign body inhalation tends to occur in younger to middle-aged animals with exposure to outdoor activities, such as hunting or running. It is much more common in dogs than in cats and can respond well to bronchoscopic removal [1,23]. Classically, foreign body pneumonia is partially responsive to antibiotic therapy, with clinical signs returning when antibiotics are discontinued. If foreign material has been present in a bronchus chronically, bronchiectasis can then occur, leading to continued or recurrent infections even after removal of foreign material. Foreign body disease can also be accompanied by pneumothorax, pyothorax, or retroperitoneal abscessation due to migration of the foreign body, and these cases typically require surgery for resolution.

Aspiration pneumonia can occur in any age dog or cat, depending on the type of underlying condition responsible for aspiration. While it tends to be a condition more commonly recognized in dogs than in cats, recent studies have highlighted the commonality of predisposing conditions in cats compared to dogs, as well as specifics of the syndrome in cats [11,13]. Laryngeal paresis or paralysis, pharyngeal dysfunction, mass lesions, or surgical interventions as an underlying cause are more common in older animals [16,31]. Therefore, in older animals with pneumonia, questions should be directed to determine the presence of voice change (dysphonia), evidence of dysphagia, or concurrent esophageal disease, which could indicate laryngeal dysfunction or mass effect.

In younger animals, aspiration pneumonia related to seizure disorders could result from portosystemic shunting, idiopathic epilepsy, or granulomatous meningoencephalitis, while brain neoplasia would be more common in middle-aged to older animals [32,33,34]. Swallowing disorders are common in dogs with aspiration pneumonia [2,31]. They can be breed-related in brachycephalic dogs, associated with megaesophagus or cricopharyngeal achalasia, or due to persistent right aortic arch in immature animals. Likewise, congenital anomalies, such as tracheoesophageal fistulas or palatal defects, can predispose neonates and juvenile animals [35]. Older animals might be affected by regurgitation or vomiting due to gastroesophageal reflux or vomiting associated with chronic enteropathies or pancreatitis.

Certain breeds of dog appear to be prone to various types of pneumonia. Overall, brachycephalic dogs have a 3.8-fold increased incidence of pneumonia, although little is known about the disease syndrome in brachycephalic cats [36]. A rhinitis/bronchopneumonia syndrome in young Irish wolfhounds, as well as recurrent bacterial pneumonia in adult Irish wolfhounds, has have been described. These conditions are suspected to arise from separate etiologies that could be aspiration-related or immune in origin, but remain poorly understood [37,38,39]. Primary ciliary dyskinesia is a rare genetic condition that can explain recurrent pneumonia in pure-bred dogs. This has been reported in a number of dog breeds, including the Old English sheepdog, Bichon Frisé, Alaskan malamute, Newfoundlands, dachshund, and many other pure-bred dogs.

## 4. Physical Examination

There are few prospective studies that have documented physical examination findings in animals with pneumonia, and it would be anticipated that different underlying causes of pneumonia would result in variable systemic manifestations. Pneumonia typically results in cough, an increased respiratory rate, and tracheal sensitivity. Detection of tachypnea can vary with the phase of disease during which the animal is examined. In a retrospective study of dogs with aspiration pneumonia, only 42% had a respiratory rate exceeding 30 breaths per minute [40], although Irish Wolfhounds with acute aspiration pneumonia commonly had respiratory rates exceeding 40 [37]. Tachypnea (respiratory rate > 40 breaths per minute) was noted in 37% of cats in one study [11].

Changes in body temperature in pneumonia are variable and normothermia is not uncommon. Approximately 30–43% of dogs with aspiration pneumonia presented with an elevated body temperature [15,40], while cats with aspiration events were either hyperthermic, consistent with a systemic inflammatory response, or hypothermic, similar to findings in cats with sepsis [11,13].

Lung sounds are variably abnormal. In retrospective studies of dogs with aspiration pneumonia, over 50% had only increased harshness to lung sounds rather than crackles or adventitious lung sounds [37,40]. Wheezing was detected in 23% of cats with bacterial pneumonia [11]. Although similar to retrospective studies in dogs, lung sounds were described only as harsh or increased in 60% of cats with aspiration [13]. Upper airway sounds, such as stridor and stertor, can be identified in animals with laryngeal dysfunction or paralysis and can impede the ability to adequately auscult an animal’s lower airways and lungs.

## 5. Differential Diagnoses

In dogs, differentials for lobar aspiration pneumonia, bronchopneumonia, or foreign body pneumonia include lung abscess, neoplasia, or embolic disease, and historical findings along with radiographic and hematologic changes can be helpful in prioritizing the differential list. In cats, the top differential for bacterial pneumonia is likely inflammatory airway disease, which closely mimics bronchopneumonia in history and radiographic findings. Interstitial fibrosis, metastatic neoplasia, and congestive heart failure can present with similar clinical findings in both species [41,42,43].

## 6. Diagnostic Testing

The diagnostic approach to each dog or cat with suspected pneumonia depends heavily on the patient’s clinical conditions, available facilities, and client factors, such as financial resources and emotional investment. In general, when suspicious of bacterial pneumonia, a comprehensive physical examination, detailed medical history, complete blood count, and 3-view thoracic radiographs are a reasonable starting point and will help determine the need for additional testing, such as advanced imaging or airway sampling.

### 6.1. Hematology

Routine laboratory testing is expected to reveal an inflammatory leukogram of varying severity, depending on the timing of blood sampling in relation to the airway injury and severity of the insult. A normal leukogram is not unusual and has been reported in 39–51% of dogs with bacterial pneumonia [15,40].

### 6.2. Acute Phase Proteins

Biomarkers for inflammation, such as acute phase proteins, have been investigated for the diagnosis and monitoring of bacterial pneumonia in dogs, but similar studies are lacking for cats. Elevations in C-reactive protein (CRP) concentrations are associated with bacterial pneumonia and normalization precedes radiographic resolution of disease, making it a more sensitive indicator of disease resolution [44,45,46]. There is a potential that elevations in CRP might precede radiographic change in peracute disease, but this has not been studied in dogs. Because these proteins are non-specific markers of inflammation, other reasons for elevations must be excluded if they are intended to be used for monitoring; however, in one study, severe elevation (>100 μg/mL) was found to have excellent specificity for bacterial pneumonia in dogs presenting with respiratory disease [47,48]. Other acute phase proteins, like serum amyloid A (SAA) and haptoglobin, have also been investigated and SAA appears sensitive to moderate to severe disease, but haptoglobin concentrations are not chronologically correlated with clinical disease or radiographic scores [44,45].

### 6.3. Immunologic Testing

Testing to evaluate for innate and acquired immunodeficiency has been described in the veterinary literature. For instance, secretory IgA deficiency was described in an Irish setter with recurrent bacterial pneumonia and concurrent gastrointestinal disease [6]. This condition is not widely recognized in veterinary medicine and serum immunoglobulin measurement might not reflect mucosal levels. Other conditions, such as trapped neutrophil syndrome and cyclic neutropenia, can be documented with commercially available PCR assays; however, their association with bacterial pneumonia is not known [49,50].

### 6.4. Pulse Oximetry and Arterial Blood Gas Analysis

Arterial blood gas analysis and, as a crude surrogate, pulse oximetry can indicate hypoxemia and be helpful in determining the severity of disease, prioritizing differential diagnoses, as well as helping to monitor response to therapy. As with biomarkers, the presence of hypoxemia is not specific to bacterial pneumonia, and other causes that result in ventilation:perfusion mismatch must be ruled out [51]. While these assessments can help guide therapy and monitor responses to treatment, they have not been evaluated in the context of outcomes.

### 6.5. Diagnostic Imaging

Three-view thoracic radiographs typically reveal an alveolar infiltrate in animals with bacterial pneumonia, although early disease can be reflected by an interstitial pattern alone [40]. The distribution of the radiographic infiltrate is important when prioritizing the type of pneumonia considered most likely and in ruling out other differentials. Aspiration pneumonia classically results in a cranioventral distribution or “middle-lung lobe infiltrates”, while an inhalational etiology would be anticipated to lead to a caudal distribution of disease [1,11]. Similarly, diseases that impair mucociliary clearance can result in a cranioventral distribution of infiltrates [4,5]. Hematogenous spread of disease would likely be widespread and diffuse, whereas inhalation of a grass awn typically results in focal or multifocal lobar infiltrates, depending on how many grass awns are inhaled. However, caution is warranted as a diffuse interstitial infiltrate was found in 1/3 of dogs with confirmed airway foreign bodies in one study [1].

Computed tomography is likely the most sensitive imaging modality for pulmonary parenchymal disease and is readily available at many specialty and emergency small animal hospitals. Ideally, images should be acquired under controlled ventilation to minimize motion artifact, atelectasis, and expiratory images. For animals in which anesthesia cannot be safely accomplished, awake or sedated CT can be performed, but the image quality will suffer due to motion artifact. Changes associated with bacterial pneumonia include hyper-attenuation with pulmonary consolidation or mosaic patterns, indicating infiltrative lung disease with concurrent air trapping [52].

Thoracic ultrasound is gaining popularity in the diagnosis of bacterial pneumonia in small animals as a rapid and relatively stress-free assessment of respiratory disease; however, studies investigating its utility for the detection of B-lines and shred signs have found variable sensitivity to and specificity for bacterial pneumonia [53,54]. Fluoroscopy can help rule out some differential diagnoses for cough, such as airway collapse, or investigate concurrent diseases, such as swallowing disorders, obstructive processes, or hiatal hernia contributing to development of pneumonia, it but does not independently establish a diagnosis [19]. Abdominal ultrasound can be used to detect evidence of delayed gastric emptying, concurrent diseases associated with ileus (e.g., pancreatitis or peritonitis), as well as structural abnormalities that can result in alterations in normal gastrointestinal transit (e.g., pyloric hypertrophy or hiatal hernia). Lastly, nuclear medicine scans can be performed to assess for gastroesophageal reflux, gastric emptying, and impaired ciliary clearance, the latter of which can imply ciliary dyskinesia [28,55].

### 6.6. Sampling

Sampling respiratory specimens from affected patients can establish a definitive diagnosis, as well as screen for risk factors for pneumonia. In many cases, these samples are collected at a single point in time, but staging of diagnostics might be indicated for animals with recurrent or complex disease. Techniques include fine needle lung aspiration, tracheal wash (endotracheal or transtracheal), and bronchoscopic bronchoalveolar lavage, with variable features available for each diagnostic test. Please see Table 1 for details. In general, the submission of cytology and aerobic, anaerobic, and Mycoplasma cultures are optimal for making a diagnosis of pneumonia [11,56,57]. Airway fluid cytology will be predominantly neutrophilic in cases with bacterial bronchopneumonia and sepsis is observed in approximately 75% of cases in dogs [56].

Fine needle lung aspiration is easy to perform and requires minimal equipment, although ultrasound guidance is recommended, along with the use of small gauge needles. It is most suitable for lobar pneumonia or for the evaluation of a nodular pattern on radiographs. The sample obtained is typically only suitable for a cytologic assessment, although with multiple aspirations, sufficient samples for a culture might be obtained. As expected, there is a risk of pneumothorax with this procedure, but this risk can be minimized with adequate sedation, use of a small gauge needle, and directed sampling of consolidated areas.

Tracheal wash can be performed to evaluate bronchial or alveolar infiltrates and the endotracheal (ETW) approach is suitable for use in small-moderate sized dogs (<15 kg), cats, puppies, and kittens. For larger dogs (>15 kg), a transtracheal wash (TTW) tends to be more appropriate and is generally more likely to provide a cellular sample; however, this technique can be more challenging to perform and is more invasive [58]. Minimal equipment is required for ETW, other than a sterile endotracheal tube and a semi-flexible sterile catheter that will reach as close to the carina as possible. Brief intravenous anesthesia sufficient to maintain intubation for 5–10 min is needed. After sterile intubation, a long catheter is passed into the trachea to the level of the 4th rib space, and 5–10 mL of sterile, non-bacteriostatic saline is slowly injected followed by 1–2 mL of air to clear the catheter. Coupage can help maximize fluid return. When available, a suction apparatus can improve the return of fluid that has been in contact with the airways. Only 1–2 mL of fluid is needed for cytology and culture, and a turbid, mucus-laden fluid is more likely to be an adequate sample for diagnosis [58]. For a transtracheal wash, a short 14–16 gauge over the needle catheter is used to penetrate between the tracheal rings; the needle is withdrawn; and a sterile 3.5 polypropylene catheter is passed to the level of the carina for the instillation and aspiration of fluid as above. Typically, only mild sedation is needed for this procedure.

Bronchoscopy allows the collection of an airway sample from a specific airway segment and is also useful for removing foreign material; however, it requires specialized general anesthesia, expensive endoscopic equipment, and a higher level of expertise to perform. The procedure to procure bronchoalveolar lavage (BAL) has been reviewed elsewhere and is beyond the scope of this review [59]. In cats, non-bronchoscopic “blind” BAL can be performed using a technique similar to that of the ETW described above, using a long flexible catheter that can reach lower airways [60]. Typically, 5–10 mL of sterile, non-bacteriostatic saline is used for these washes in cats, and suction should be applied using a syringe.

The cytology of an airway sample from a dog or cat with pneumonia typically reveals neutrophilic inflammation often characterized by degenerate neutrophils, and tracheal wash samples with visible intracellular bacteria were more likely to have positive cultures [58]. However, in dogs with lower respiratory tract infections, 25% of cases can have infection documented via culture, despite an absence of intracellular bacteria on cytology, highlighting that bacterial cultures should be submitted despite cytologic findings when pneumonia is suspected [56].

The interpretation of bacterial cultures from an airway sample must consider cytologic characteristics, as well as the normal microbiologic inhabitants of the lower airways. Samples that have oropharyngeal contaminants (squamous cells or Conchiformibius bacteria—previously called Simonsiella) can be anticipated to have light growth of aerobes, anaerobes, and Mycoplasma, which normally colonize the upper airway. Unfortunately, these bacteria can also be responsible for pneumonia, which is confirmed by the presence of septic inflammation with intracellular bacteria and degenerate neutrophils.

Standard microbiologic culture remains the preferred method of organism detection, but molecular techniques, like targeted PCR assays and broad-spectrum organism sequencing of BAL samples, have shed new insight into the complex ecosystem within the lower respiratory tract. Unfortunately, PCR on oropharyngeal swabs rarely provides clinically relevant information. Similarly, bacterial blood culture does not appear to be an adequate surrogate for airway lavage cytology and culture [61].

In a young animal with recurrent, antibiotic-responsive pneumonia, as mentioned above, an important differential diagnosis to investigate is primary ciliary dyskinesia. In an intact male that is sexually mature, the simplest test to rule out ciliary dyskinesia is sperm motility, because all cilia in the body are affected by the dynein arm abnormality. Forward and purposeful sperm motion would rule out ciliary dyskinesia. Electron microscopy can also be used to evaluate ciliary structure in a biopsy sample of respiratory epithelium [28]. Because cilia are present throughout the respiratory tract, specimens from the nasal passage are typically sampled as these are easier to acquire than other airway biopsies. As a specialized diagnostic test, it is important to identify a laboratory capable of interpreting these biopsies, which must be preserved in appropriate fixative.

## 7. Conclusions

Pneumonia remains one of the most common diagnoses of dogs and cats that present to veterinarians with respiratory signs. Animals with a history of vomiting, regurgitation, recent anesthesia, or laryngeal or pharyngeal disease should be assessed for the presence of aspiration pneumonia. The diagnostic approach to each patient will vary based on signalment, history, examination, clinical findings, and available resources, but generally should include hematology assessment, diagnostic imaging, and, potentially, airway sampling to identify the cause of disease and guide the therapeutic plan. New diagnostic options, such as lung ultrasound and molecular testing, may help streamline diagnosis and these deserve further study.

## Figures and Tables

**Table 1 animals-14-01601-t001:** Characteristics and requirements of respiratory tract sampling techniques.

	Ultrasound-Guided FNA	Transtracheal Wash	Endotracheal Wash	Bronchoscopy	Biopsy
**Sedation or anesthesia**	Heavy sedation to control unwanted movement.	Light sedation to allow catheter placement.	Brief IV anesthesia to allow intubation.	General anesthesia required.	General anesthesia required.
**Equipment**	Ultrasound; 25, 26, 27 gauge needles.	Large (14, 16 gauge) catheter, long collection catheter, suction apparatus.	Sterile endotracheal tube, long collection catheter, suction apparatus.	Flexible endoscope.	Rigid or flexible endoscope, cup biopsy forceps.
**Risks and complications**	Pneumothorax, hemorrhage.	Pain, hemorrhage, emphysema.	Pneumomediastinum, baro-trauma.	Apnea, cardiorespiratory arrest.	Hemorrhage.
**Sample characteristics**	Focal sample. Usually can only obtain cytology.	Carinal sample. Cytology and culture.	Carinal sample. Cytology and culture. Prone to upper airway/oral contamination.	Alveolar segment. Cytology and culture.	Mucosal/submucosal respiratory epithelium. Histopathology.
**Indication**	Evaluation of pulmonary consolidation.	Less invasive and resource intensive airway fluid collection. Generally limited to cooperative, large dogs.	Minimally resource intensive airway fluid collection. Can be performed on all animals.	Visual inspection & airway fluid collection.	Evaluation of endobronchial masses & collection of tissue for EM.

## Data Availability

No new data were created or analyzed in this study. Data sharing is not applicable to this article.

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
