# Peer review of "Recognition and Diagnosis of Underlying Disease Processes in Bacterial Pneumonia"

_animals, 2024, doi:10.3390/ani14111601_

Round 1

Reviewer 1 Report

Comments and Suggestions for Authors

A review article entitled ”Recognition and diagnosis of underlying disease processes in bacterial pneumonia” presents a comprehensive summary of the current research information concerning possible etiologies of bacterial pneumonia in dogs and cats. The review is well written and as such should be published. I have a few minor issues I would like to suggest:

The headline indicates that both “recognition and diagnosis of underlying disease processes” would be discussed in the manuscript. The beginning of the manuscript (Section 2. Definition and etiology) comprehensively introduces various possible etiologies for bacterial pneumonia, but the sections 4. (Physical examination) and 6. (Diagnostic testing) focus almost solely in the clinical signs and diagnostic testing of bacterial pneumonia in general regardless of etiology. The authors could consider either modifying the title of the manuscript to better describe the content or including more information on possible specific testing for different etiologies in addition to the information already provided concerning diagnostic testing for ciliary dyskinesia (eg. potential biomarkers for aspiration, laryngeal/esophageal function testing, diagnostic imaging for structural changes, testing for immune dysfunction, testing for viral etiology etc.).

It could also be discussed, when general diagnostics for bacterial pneumonia (sampling for cytology and bacterial culture for example) are indicated and when the clinician should pursue advanced diagnostics in order to identify the underlying etiology.

There is a relatively loose way of providing references with several statements in the manuscript presenting information from previous research work, but not providing the reference to the study. I am not sure whether this is requested by the paper?

Lines 84-86: Also, chronic bronchial inflammation (eg. chronic bronchitis or eosinophilic bronchitis) may cause bronchiectasis development.

Lines 173-175: This sentence could be specified a bit: In Irish Wolfhounds both a rhinitis/bronchopneumonia syndrome of puppies/young dogs as well as an increased incidence of bacterial pneumonia and recurrent episodes of bacterial pneumonia in adult dogs have been described. Due to differences in the clinical picture, it is likely that these are not the same. Etiology of these is unknown and both immune deficit and aspiration etiology have been suggested.

Line 178: The authors could add congenital and acquired broncho- or tracheoesopgaheal fistulas as a possible etiology for recurring bacterial pneumonia.

Lines 188-189: The paper describing recurrent pneumonia in Irish Wolfhounds examined the affected dogs in between pneumonia episodes – the findings in this study do not represent clinical signs during pneumonia.

Lines 200-201: Same comment as lines 188-189.

Lines 244-254: Is there any information on how well thoracic radiographs can indicate etiology? Cranioventral distribution may be common also in pneumonias not related to aspiration (for example bronchiectasis is common in these areas, also many ciliary dyskinesia patients develop pneumonia in these lobes).

Line 266: Please comment also on the ability of US to differentiate between etiologies of bacterial pneumonia.

Lines 285-292: Do the authors use fine needle aspirates often when suspecting bacterial pneumonia? How do the authors see the diagnostic yield vs. risk of pneumothorax for this technique? If this is not a diagnostic technique often used, this could be stated.

Table 1: Transtracheal wash sampling could be added to the table as it is different when it comes to risks and benefits when compared to ETW.

Author Response

Reviewer 1

<REV1>A review article entitled ”Recognition and diagnosis of underlying disease processes in bacterial pneumonia” presents a comprehensive summary of the current research information concerning possible etiologies of bacterial pneumonia in dogs and cats. The review is well written and as such should be published. I have a few minor issues I would like to suggest: 

<AU>Thank you for your words and thoughtful comments.

<REV1>The headline indicates that both “recognition and diagnosis of underlying disease processes” would be discussed in the manuscript. The beginning of the manuscript (Section 2. Definition and etiology) comprehensively introduces various possible etiologies for bacterial pneumonia, but the sections 4. (Physical examination) and 6. (Diagnostic testing) focus almost solely in the clinical signs and diagnostic testing of bacterial pneumonia in general regardless of etiology. The authors could consider either modifying the title of the manuscript to better describe the content or including more information on possible specific testing for different etiologies in addition to the information already provided concerning diagnostic testing for ciliary dyskinesia (eg. potential biomarkers for aspiration, laryngeal/esophageal function testing, diagnostic imaging for structural changes, testing for immune dysfunction, testing for viral etiology etc.). 

<AU> Additions to the physical examination and diagnostic testing sections have been made to reflect the PE and work up of animals with laryngeal disease, GI dysmotility, anatomic risk factors for recurrent disease and immunodeficiencies.

<REV1>It could also be discussed, when general diagnostics for bacterial pneumonia (sampling for cytology and bacterial culture for example) are indicated and when the clinician should pursue advanced diagnostics in order to identify the underlying etiology. 

<AU>A preamble to the diagnostic testing section (6) has been added with the authors’ opinion on this.

<REV1>There is a relatively loose way of providing references with several statements in the manuscript presenting information from previous research work, but not providing the reference to the study. I am not sure whether this is requested by the paper?

<AU>We have tried to provide citations when appropriate, as requested by reviewer 3.

<REV1>Lines 84-86: Also, chronic bronchial inflammation (eg. chronic bronchitis or eosinophilic bronchitis) may cause bronchiectasis development.

<AU>We have added inflammatory airway disease to the list of comorbidities that can result in bronchiectasis.

<REV1>Lines 173-175: This sentence could be specified a bit: In Irish Wolfhounds both a rhinitis/bronchopneumonia syndrome of puppies/young dogs as well as an increased incidence of bacterial pneumonia and recurrent episodes of bacterial pneumonia in adult dogs have been described. Due to differences in the clinical picture, it is likely that these are not the same. Etiology of these is unknown and both immune deficit and aspiration etiology have been suggested. 

<AU>We have edited this section to provide more detail.

<REV1>Line 178: The authors could add congenital and acquired broncho- or tracheoesopgaheal fistulas as a possible etiology for recurring bacterial pneumonia. 

<AU>Added.

<REV1>Lines 188-189: The paper describing recurrent pneumonia in Irish Wolfhounds examined the affected dogs in between pneumonia episodes – the findings in this study do not represent clinical signs during pneumonia. 

<AU>The reference to Irish wolfhounds with recurrent pneumonia between episodes of disease has been removed.

<REV1>Lines 200-201: Same comment as lines 188-189.

<AU>Removed, as above.

<REV1>Lines 244-254: Is there any information on how well thoracic radiographs can indicate etiology? Cranioventral distribution may be common also in pneumonias not related to aspiration (for example bronchiectasis is common in these areas, also many ciliary dyskinesia patients develop pneumonia in these lobes). 

<AU>There are descriptive studies that suggest that aspiration pneumonia (and impaired clearance, as this reviewer reminds us) tend to have cranioventral distribution (Kogan, Tart, Greenwell, Dear) while other etiologies more often present with variable radiographic patterns (Tenwolde), but to these authors’ knowledge there aren’t hypothesis driven studies determining the accuracy of predicting etiology of pneumonia based on radiographic findings. This study would undoubtably be very difficult to perform as the determination of etiology is a Catch 22 in that the clinician often relies on the distribution of pulmonary infiltrates to determine likely cause.

<REV1>Line 266: Please comment also on the ability of US to differentiate between etiologies of bacterial pneumonia. 

<AU>To the authors’ knowledge, this has not been evaluated and we feel that the statement mentioning variable sensitivity and specificity of the test adequately addresses the topic.

<REV1>Lines 285-292: Do the authors use fine needle aspirates often when suspecting bacterial pneumonia? How do the authors see the diagnostic yield vs. risk of pneumothorax for this technique? If this is not a diagnostic technique often used, this could be stated. 

<AU>We do not routinely perform lung FNA for cases suspected of bacterial pneumonia but might consider it to investigate differentials such as pulmonary neoplasia or granulomas that can sometimes result in similar radiographic patterns. We have added a row entitled ‘Indication’ to address this inquiry.

<REV1>Table 1: Transtracheal wash sampling could be added to the table as it is different when it comes to risks and benefits when compared to ETW.

<AU>This has been added.

Reviewer 2 Report

Comments and Suggestions for Authors

It is an interesting work that highlights the importance of thorough clinical investigation to identify primary pathologies capable of predisposing dogs and cats to bacterial pneumonia.

Few points need clarification:

Line 47: How does ciliary dyskinesia cause local immunodeficiency?

Line 75: How does laryngeal dysfunction cause aspiration? Laryngeal dysfunction alone, unassociated with pharyngeal dysfunction, very rarely causes aspiration pneumonia.

Line 77: If the dog has arytenoid paralysis and is concurrently dysphagic due to pharyngeal dysfunction, the risk of aspiration is high; conversely, in the presence of arytenoid paralysis alone, the risk is negligible.

Line 79: Among predisposing causes in brachycephalic dogs, I would add "aberrant pharyngeal anatomy"; additionally, tracheal hypoplasia, relatively common in brachycephalic dogs, causes severe pediatric forms of bronchopneumonia.

Line 156: I would add "pharyngeal dysfunctions".

Lines 166-167: In brachycephalic breeds, cricopharyngeal achalasia is uncommon based on articles published on the topic; furthermore, this condition should not be included in esophageal pathologies as it pertains to the pharynx.

Line 313: How should anesthesia for bronchoscopic examination be "specialized"?

Line 315: "To add: 'In cats, non-bronchoscopic blind BAL...'"

Line 322: To add: "...with intra-cellular visible bacteria..."

Line 354: to be changed: "...laryngo-pharyngeal disease should be..."

In many paragraphs of the article, statements are made without pertinent bibliographic references; I would insert pertinent bibliographic references at lines 50, 78, 104, 127, 178, 250, 262, 343.

Line 354: "...laryngopharyngeal disease should be..."

Author Response

Reviewer 2

<REV2>It is an interesting work that highlights the importance of thorough clinical investigation to identify primary pathologies capable of predisposing dogs and cats to bacterial pneumonia.

<AU>Thank you for your thoughtful peer review of our manuscript.

<REV2>Few points need clarification:

<REV2>Line 47: How does ciliary dyskinesia cause local immunodeficiency?

<AU>Ciliary clearance via the mucociliary apparatus is a feature of innate, mucosal immunity and impaired mucociliary clearance results in entrapment of foreign material within the lungs and airways, resulting in impaired innate immunity (Marshall JS, Warrington R, Watson W, et al. An introduction to immunology and immunopathology. Allergy Asthma Clin Immunol 2018;14:49.)

<REV2>Line 75: How does laryngeal dysfunction cause aspiration? Laryngeal dysfunction alone, unassociated with pharyngeal dysfunction, very rarely causes aspiration pneumonia.

<AU>Studies have identified laryngeal dysfunction and/or laryngeal lateralization as a major risk factor for aspiration pneumonia. In some cases animals in these studies have identifiable esophageal dysmotility but in many cases no such concurrent disease is found. (Wilson D, Monnet E. Risk factors for the development of aspiration pneumonia after unilateral arytenoid lateralization in dogs with laryngeal paralysis: 232 cases (1987–2012). Journal of the American Veterinary Medical Association 2016;248:188-194; Taylor SS, Harvey AM, Barr FJ, et al. Laryngeal disease in cats: a retrospective study of 35 cases. Journal of feline medicine and surgery 2009;11:954-962.; Kogan DA, Johnson LR, Sturges BK, et al. Etiology and clinical outcome in dogs with aspiration pneumonia: 88 cases (2004-2006). Journal of the American Veterinary Medical Association 2008;233:1748-1755.)

<REV2>Line 77: If the dog has arytenoid paralysis and is concurrently dysphagic due to pharyngeal dysfunction, the risk of aspiration is high; conversely, in the presence of arytenoid paralysis alone, the risk is negligible.

<AU>The authors dispute this assertion and would prefer to leave the sentence as written.

<REV2>Line 79: Among predisposing causes in brachycephalic dogs, I would add "aberrant pharyngeal anatomy"; additionally, tracheal hypoplasia, relatively common in brachycephalic dogs, causes severe pediatric forms of bronchopneumonia.

<AU>This has been added.

<REV2>Line 156: I would add "pharyngeal dysfunctions".

<AU>Added.

<REV2>Lines 166-167: In brachycephalic breeds, cricopharyngeal achalasia is uncommon based on articles published on the topic; furthermore, this condition should not be included in esophageal pathologies as it pertains to the pharynx.

<AU>This sentence was rewritten to broaden to swallowing disorders rather than esophageal dysfunction.

<REV2>Line 313: How should anesthesia for bronchoscopic examination be "specialized"?

<AU>In order to optimize ventilation and adequate oxygen delivery during the procedure specialized equipment such as jet ventilators are often necessary for small animals in particular where the bronchoscope would either occlude the entire lumen of the endotracheal tube or be too fragile to safely pass through ET tube adapters.

<REV2>Line 315: "To add: 'In cats, non-bronchoscopic blind BAL...'"

<AU>Added.

<REV2>Line 322: To add: "...with intra-cellular visible bacteria..."

<AU>Added.

<REV2>Line 354: to be changed: "...laryngo-pharyngeal disease should be..."

<AU>Amended.

<REV2>In many paragraphs of the article, statements are made without pertinent bibliographic references; I would insert pertinent bibliographic references at lines 50, 78, 104, 127, 178, 250, 262, 343.

<AU>

<REV2>Line 354: "...laryngopharyngeal disease should be..."

<AU>Amended, as above.

Reviewer 3 Report

Comments and Suggestions for Authors

I would like to thank the authors for this review. This manuscript provides a good review of historical information and summarizes this collected data well. However, there are some additions from the current literature that I think are needed prior to publication.

Including some comments about the diagnostic criteria for bacterial infection and the need for antimicrobial therapy in veterinary patients is needed. There have been a few recent papers that have discussed this issue (Howard et al. 2021, Lebastard et al 2022) among others. As well as several others that discuss management of pneumonia without antimicrobials. As a departure from the historical norm, this seems like an important point of discussion for a manuscript of this type. Additionally, given the discussion aspiration within the manuscript it seems a more extensive section of diagnosis of swallow dysfunction and aspiration syndromes seems relevant.  A table may be helpful to outline the number of established conditions discussed within the manuscript. Finally, there are numerous statements that are made by the authors that require citations. It was confirmed with the editor that there is no limit to the number citations for the current manuscript. Therefore, the authors should provide adequate citations for these statements throughout the manuscript. A list of some of the statements lacking clear citations is provided below.  

Line 39: Citation needed.

Line 46: Citation needed.

Line 48: Citation needed.

Line 50: Citation needed.

Line 56: Citation needed.

Line 58: Citation needed.

Line 62: Citation needed.

Line 70: Citation needed.

Line 81: Citation needed.

Line 84: Citation needed.

Line 86: Citation needed.

Line 97: Citation needed.

Line 103: Citation needed.

Line 107: Citation needed.

Line 133: Citation needed.

Line 157: Citation needed.

Line 164: Citation needed.

Line 212: Citation needed.

Line 234: Citation needed.

Line 246: Citation needed.

Line 250: Citation needed.

Line 262: Citation needed.

Line 269: Citation needed.

Line 271: Citation needed. Additionally, scintigraphy has recently been used to evaluate reflux in dogs.

Thank you again for your time and effort in generating this manuscript and I look forward to reading an edited version.

Author Response

Reviewer 3

<REV3>I would like to thank the authors for this review. This manuscript provides a good review of historical information and summarizes this collected data well. However, there are some additions from the current literature that I think are needed prior to publication.

<AU>Thank you for your thoughtful review of our manuscript and suggested amendments.

<REV3>Including some comments about the diagnostic criteria for bacterial infection and the need for antimicrobial therapy in veterinary patients is needed. There have been a few recent papers that have discussed this issue (Howard et al. 2021, Lebastard et al 2022) among others. As well as several others that discuss management of pneumonia without antimicrobials. As a departure from the historical norm, this seems like an important point of discussion for a manuscript of this type.

<AU>We agree that this is an important area to discuss in the context or management and treatment of bacterial pneumonia, however it is outside the scope of this work which is focused on identification and diagnosis of the disease instead.

<REV3>Additionally, given the discussion aspiration within the manuscript it seems a more extensive section of diagnosis of swallow dysfunction and aspiration syndromes seems relevant.  A table may be helpful to outline the number of established conditions discussed within the manuscript.

<AU>As a manuscript focused broadly on bacterial pneumonia, we feel that a table devoted to aerodigestive disorders isn’t necessary for the work. We will defer to the editor here if they would like the table added.

<REV3>Finally, there are numerous statements that are made by the authors that require citations. It was confirmed with the editor that there is no limit to the number citations for the current manuscript. Therefore, the authors should provide adequate citations for these statements throughout the manuscript. A list of some of the statements lacking clear citations is provided below.  

<AU>We have added relevant citations as recommended.

<REV3>Line 39: Citation needed.

<AU>Several studies exist that describe incidence of bacterial pneumonia in dogs and cats but not in the overall context of ranking it amongst other pathogens, this is why we state that bacterial pneumonia is likely the most common.

<REV3>Line 46: Citation needed.

<AU>Added.

<REV3>Line 48: Citation needed.

<AU>Added.

<REV3>Line 50: Citation needed.

<AU>Added.

<REV3>Line 56: Citation needed.

<AU>It is the authors’ experience that use of glucocorticoids do not increase risk of bacterial pneumonia in dogs and cats. Glucocorticoids are the cornerstone of therapy for most inflammatory airway diseases and studies and case reports discuss their use without mentioning increased risk for development of bacterial infection. Furthermore, risk factors for development bacterial pneumonia in dogs and cats have been studied and none of these studies identified glucocorticoid use as a risk factor.

<REV3>Line 58: Citation needed.

<AU>Added.

<REV3>Line 62: Citation needed.

<AU>Added.

<REV3>Line 70: Citation needed.

<AU>Added

<REV3>Line 81: Citation needed.

<AU>Added.

<REV3>Line 84: Citation needed.

<AU>Added.

<REV3>Line 86: Citation needed.

<AU>Added.

<REV3>Line 97: Citation needed.

<AU>Added and amended.

<REV3>Line 103: Citation needed.

<AU>Added.

<REV3>Line 107: Citation needed.

<AU>Added.

<REV3>Line 133: Citation needed.

<AU>Added.

<REV3>Line 157: Citation needed.

<AU>Added.

<REV3>Line 164: Citation needed.

<AU>Added.

<REV3>Line 212: Citation needed.

<AU>Added.

<REV3>Line 234: Citation needed.

<AU>Added.

<REV3>Line 246: Citation needed.

<AU>Added.

<REV3>Line 250: Citation needed.

<AU>Added.

<REV3>Line 262: Citation needed.

<AU>Added.

<REV3>Line 269: Citation needed.

<AU>Added.

<REV3>Line 271: Citation needed. Additionally, scintigraphy has recently been used to evaluate reflux in dogs.

<AU>Added. Thank you.

<REV3>Thank you again for your time and effort in generating this manuscript and I look forward to reading an edited version.

<AU>Thank you for your efforts in helping to refine this manuscript!